# PlantaNet and PlantaNetLite: Efficient and explainable multi-crop plant disease classification via transformer benchmarking and custom lightweight CNNs

Md. Sifat Haque Zidan[ID], Al Imran[ID], Md. Kayes Mia[ID], Muhammad Hussain, Ahmed Faizul Haque Dhrubo[ID]*, Mohammad Abdul Qayum

Department of Electrical and Computer Engineering, North South University, Dhaka, Bangladesh

* ahmed.dhrubo@northsouth.edu

## Abstract

Plant disease diagnosis based on visual symptoms is crucial for preventing yield loss; however, deployment in practical settings remains challenging due to inter-class similarity, background noise, and limited computational resources. This study presents a plant disease classification framework evaluated on a curated multi-crop dataset aggregated from multiple publicly available repositories, comprising 51 disease and healthy classes. The dataset includes approximately 45,000 original images that were expanded through controlled augmentation during training to improve generalization. We benchmark eight ImageNet-pretrained tiny vision transformer architectures trained for up to 50 epochs. Among these, CAFormer-s18 achieved strong validation performance but with increased computational overhead. To enable efficient and computationally lightweight solutions, we design two fully customized convolutional neural networks: PlantaNetLite (1.28M parameters) and PlantaNet (2.58M parameters). After hyperparameter optimization and full 100-epoch training, PlantaNet achieved 99.37% validation accuracy and 99.66% test accuracy with a compact model size (9.85 MB) and moderate computational cost, while PlantaNetLite achieved a best validation accuracy of 99.22% under further parameter reduction. Qualitative Grad-CAM and Grad-CAM++ analyses provide insight into the regions influencing model predictions. Overall, the proposed models demonstrate competitive accuracy while maintaining computational efficiency, highlighting their potential suitability for resource-constrained deployment scenarios.

## 1 Introduction

Plant diseases cause substantial yield and quality losses worldwide, particularly in staple and high-value crops. Early and accurate diagnosis is critical, as many diseases spread rapidly and exhibit visible symptoms only after infection has

**Data availability statement:** The dataset used in this study is publicly available at https://www.kaggle.com/datasets/alimransonet/plant-disease-dataset.

**Funding:** The author(s) received no specific funding for this work.

**Competing interests:** The authors have declared that no competing interests exist.

progressed. In resource-limited agricultural environments, disease identification still relies heavily on expert inspection, which is time-consuming, subjective, and often unavailable at scale. Consequently, automated plant disease recognition from leaf images has emerged as a key research direction for advancing precision agriculture and sustainable crop management. Recent surveys indicate that computer vision–based disease detection has made significant progress, with deep learning techniques dominating current solutions due to their ability to learn discriminative visual representations directly from data [1,2].

Traditional machine-learning approaches for plant disease diagnosis relied on handcrafted features, such as color, texture, and shape descriptors, followed by classical classifiers. These methods are highly sensitive to variations in illumination, background clutter, scale, and intra-class symptom diversity. Deep convolutional neural networks (CNNs) address many of these limitations by hierarchically extracting robust visual features and have demonstrated strong performance on benchmark datasets, including PlantVillage and real-field collections [3,4]. However, despite their high accuracy, many CNN-based systems remain computationally intensive and difficult to deploy on edge devices. In addition, their decision-making processes are often opaque to agronomists and farmers. These limitations highlight two persistent challenges in the literature: (i) the need for lightweight, high-accuracy models suitable for practical deployment, and (ii) the integration of explainable artificial intelligence (XAI) techniques to improve model transparency and trust [1,2].

Fig 1 illustrates the motivation behind adopting a lightweight and explainable deep learning approach for plant disease diagnosis.

In parallel with CNN advancements, transformer-based vision models have emerged as competitive alternatives. Vision Transformers (ViTs) and their compact variants are capable of modeling long-range dependencies and global contextual information, which can be beneficial for recognizing subtle disease patterns distributed across leaf surfaces. Several recent studies report that transformer architectures can match or outperform CNN baselines for plant disease classification when sufficient training data and strong augmentation strategies are available [5–9]. Nevertheless, transformer-based models typically incur higher computational costs and may exhibit degraded performance under limited training data or domain-shift conditions. This motivates continued investigation into efficient CNN-based and hybrid architectures that balance classification accuracy with deployment feasibility [1,2].

Another critical challenge in plant disease recognition is dataset realism. Many publicly available datasets consist of laboratory-style images with uniform backgrounds, whereas real-world field images are affected by occlusion, mixed illumination, overlapping leaves, and complex background textures. To reduce this gap, recent research emphasizes robust data augmentation strategies and large-scale multi-crop datasets to improve generalization from controlled environments to real agricultural fields [1,2,10] (Fig 2).

Beyond predictive performance, interpretability has become increasingly important for agricultural AI systems. For real-world adoption, models must demonstrate that their predictions are based on biologically meaningful disease symptoms rather than

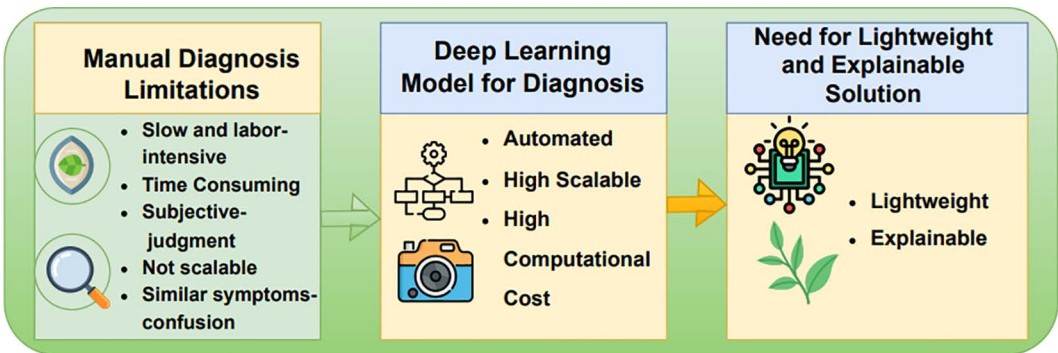

**Fig 1. Motivation and need for a lightweight and explainable deep learning–based plant disease diagnosis framework.**

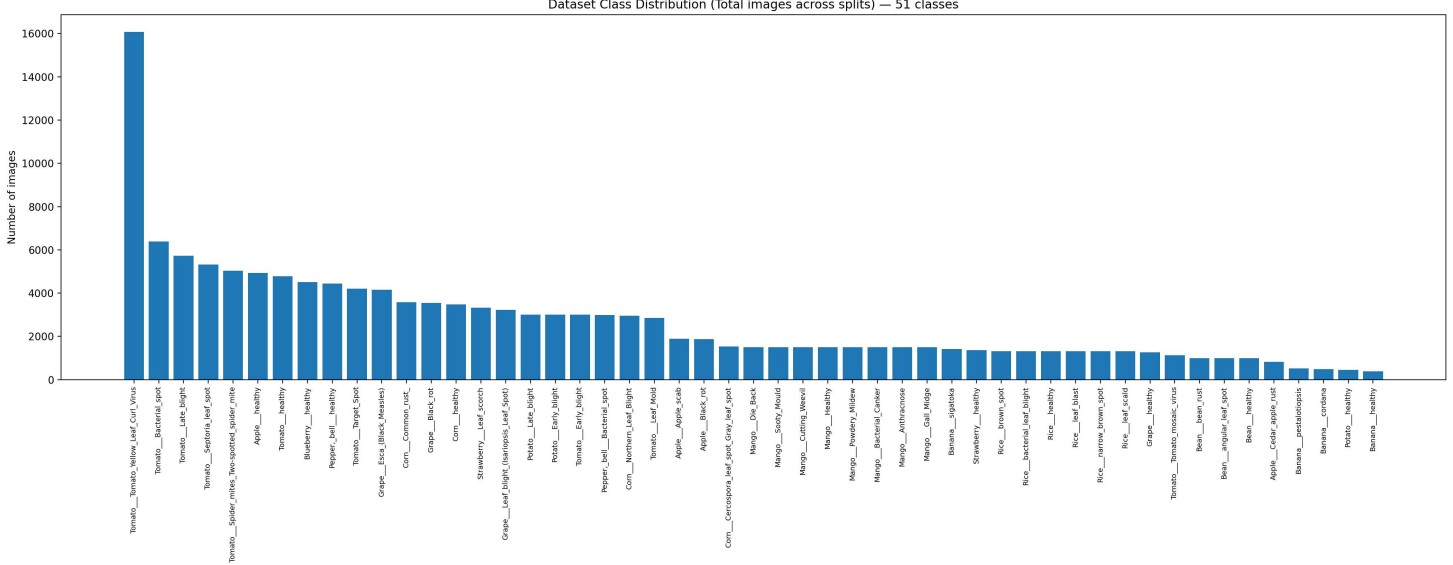

**Fig 2. Dataset class distribution(total images across splits)-51 classes.**

spurious background artifacts. Gradient-based localization methods, such as Grad-CAM and Grad-CAM++, are widely used explainability techniques that generate visual heatmaps highlighting influential regions in input images. Employing both methods provides complementary qualitative perspectives on model attention patterns, offering additional insight into prediction behavior [11].

## 1.1 Research motivation and contributions

Despite extensive research, several key limitations persist in existing plant disease classification studies. First, many high-performing models exhibit a significant accuracy–efficiency trade-off, making them unsuitable for deployment in resource-constrained environments [12,13]. Second, a large portion of the literature evaluates either CNNs or transformer-based models in isolation, without systematic cross-family comparisons under consistent experimental settings [1,2]. Third, explainability is often presented as auxiliary visualization without systematic analysis of model attention behavior [11,14]. A comparison of the proposed workflow with typical prior work is provided in Table 1.

**Table 1. Comparison of the proposed pipeline with typical prior work.**

| Study Type | Multi-crop (10 + crops) | Large-scale (post-augmentation) | Tiny transformer baselines | Lightweight CNN (≤3M params) | Hyperparameter tuning | Grad-CAM | Grad-CAM++ |
|---|---|---|---|---|---|---|---|
| Typical CNN-only papers | Yes / No | No / Yes | No | No / Yes | No / Yes | Yes | No |
| Typical ViT-only papers | Yes / No | No / Yes | Yes | No | No / Yes | Yes / No | No / Yes |
| This work | Yes | Yes | Yes (8 models) | Yes (1.28M, 2.58M) | Yes (multi-stage tuning) | Yes | Yes |

To address these limitations, this paper makes the following contributions:

- We curate and preprocess a multi-crop plant disease dataset aggregated from multiple publicly available repositories, comprising 51 classes and expanded through controlled augmentation applied to the training subset to improve robustness and generalization [10].

- We conduct a systematic benchmark of eight compact vision transformer architectures trained under identical conditions to establish strong and fair attention-based baselines.

- We propose two fully customized lightweight CNN architectures, PlantaNetLite (1.28M parameters) and PlantaNet (2.58M parameters), designed specifically for efficient and accurate plant disease recognition [13,15].

- We perform hyperparameter-driven optimization using learning rate, weight decay, and MixUp regularization to identify configurations that improve validation performance [16].

- We analyze model attention patterns using Grad-CAM and Grad-CAM++ to qualitatively examine the image regions influencing predictions [11,14].

- We conduct controlled ablation experiments to quantify the contribution of architectural components including normalization strategy, dropout regularization, and additional convolutional refinement blocks.

Detailed experimental analysis and ablation results are presented in Section 4 to further examine the contribution of individual architectural components. Given their compact sizes and competitive validation performance, PlantaNetLite and PlantaNet demonstrate potential suitability for resource-constrained deployment scenarios. PlantaNetLite is well suited for mobile and edge-based environments with strict memory constraints, while PlantaNet offers a balanced accuracy–efficiency trade-off for settings with moderately higher computational capacity. Further validation on independently collected field datasets would strengthen assessment of real-world robustness.

## 2 Related work

Early research on image-based plant disease recognition relied on handcrafted features (e.g., color histograms, texture operators, and shape descriptors) combined with classical classifiers such as SVM and Random Forest. Although these approaches are computationally efficient, their performance often degrades under real-field variations including illumination changes, complex backgrounds, and partial occlusions, which limits practical deployment. The transition to deep learning improved robustness by enabling feature learning directly from raw images. A landmark study by Mohanty *et al.* demonstrated the effectiveness of transfer-learned CNNs on PlantVillage, establishing a strong baseline for modern plant disease classification [3].

### 2.1 CNN-based plant disease classification

CNNs remain widely adopted due to their strong inductive bias for local pattern learning, which is well-suited for recognizing lesions, spots, mildew, and texture-based symptoms. Numerous studies from 2023–2025 report improved accuracy

across crops such as rice, tomato, grape, apple, and maize using deeper backbones (e.g., ResNet, DenseNet, Efficient-Net) and task-specific refinements [1,2,4,12]. Prior work also indicates that well-tuned CNNs can outperform transformer variants in visually subtle plant disease settings, as convolutional filters naturally emphasize localized symptom cues [1,2].

Lightweight and mobile CNNs represent another important direction. Comparative studies of compact architectures such as MobileNet and ShuffleNet demonstrate that high accuracy can be maintained under reduced parameter budgets, enabling on-device inference in low-resource farming contexts [12,13]. This motivates our design of PlantaNetLite (1.28M parameters) and PlantaNet (2.58M parameters), which target a favorable balance between accuracy and efficiency.

## 2.2 Vision transformers in plant disease recognition

Recently, Vision Transformers (ViTs) and compact transformer families have been explored for plant disease recognition due to their ability to model long-range dependencies and global context, which can be useful when symptoms span distributed regions of a leaf. Architectures such as PVT, PiT, ViT-Tiny, and XCiT have shown promising results in agriculture-oriented classification tasks [5–9]. However, transformer performance is often dataset-sensitive. Multiple studies report that ViTs typically require large-scale pretraining and/or strong augmentation strategies to generalize competitively in plant disease settings [1,2,5]. These findings align with our empirical observation that several tiny transformer baselines achieve high validation accuracy, while the proposed CNNs demonstrate competitive performance with reduced computational cost under extended training schedules.

## 2.3 Hybrid and efficient architectures

To balance accuracy and efficiency, hybrid CNN–transformer networks have been proposed. Common approaches include inserting attention modules within CNN stages or using CNN stems followed by lightweight transformer blocks to combine locality with global context modeling [1,2]. In parallel, architectural efficiency techniques such as depthwise-separable convolution, squeeze-and-excitation, group normalization, and dropout scheduling are frequently used to reduce redundancy while maintaining performance [13,15]. The proposed PlantaNet family follows these principles through depthwise-separable blocks, GroupNorm-based stabilization, and progressive dropout.

## 2.4 Data augmentation and regularization strategies

Plant disease datasets often exhibit class imbalance, background bias, and subtle inter-class differences. As a result, strong augmentation (e.g., random cropping, rotation, color jitter, blur) is commonly applied to improve robustness [1,2,10]. Beyond standard augmentation, MixUp and CutMix regularization strategies have shown measurable benefits in fine-grained agricultural classification by improving decision boundary smoothness and robustness to label noise [16,17]. Consistent with these findings, our training pipeline applies geometric and photometric augmentation strategies and integrates MixUp regularization, where $\alpha$ = 0.2 provided improved validation performance during hyperparameter tuning.

## 2.5 Explainability in plant disease models

Explainability is increasingly required for real-world agricultural adoption, as farmers and agronomists need to verify whether a model focuses on disease regions rather than background artifacts. Grad-CAM and Grad-CAM++ remain among the most widely used visualization methods because they are architecture-agnostic and produce intuitive heatmaps highlighting influential regions [11,14]. Prior studies report that (i) CNN-based attention maps are often sharper and more lesion-localized than ViT-based maps due to spatial locality in convolutional layers [1,2], and (ii) Grad-CAM++ can provide finer localization than Grad-CAM for multi-lesion or scattered symptom patterns by leveraging higher-order gradient information [11].

## 2.6 Summary of research gap

From the above review, three gaps remain evident:

- **Accuracy–efficiency trade-off:** Many studies rely on heavy CNN/ViT backbones that are unsuitable for deployment, while lightweight models often suffer accuracy degradation on multi-crop, multi-disease benchmarks [5,12,13].

- **Limited evaluation across diverse crops:** Several works focus on single-crop datasets or a small number of classes, which can inflate reported performance compared to realistic multi-crop settings [1,2].

- **Explainability assessment limitations: Many studies provide visual explanations without systematic cross-class comparison or complementary localization analysis** [11,14].

Our work addresses these gaps by proposing two fully customized lightweight CNNs (PlantaNetLite and PlantaNet), benchmarking against multiple tiny transformer baselines under consistent settings, conducting hyperparameter tuning with MixUp regularization, and providing qualitative Grad-CAM/Grad-CAM++ analysis to examine model attention behavior within a computationally efficient framework.

## 3 Methodology

This section presents the complete experimental pipeline adopted in this study, encompassing dataset construction, pre-processing and augmentation, baseline benchmarking with lightweight vision transformers, the design of two customized convolutional neural networks (PlantaNetLite and PlantaNet), hyperparameter tuning, final training and evaluation protocols, and post-hoc explainability using Grad-CAM and Grad-CAM++.

### 3.1 Overall Framework

The proposed workflow follows a structured six-stage pipeline:

- Dataset compilation and stratified split preparation

- Training subset augmentation

- Baseline benchmarking using tiny vision transformers

- Design and development of customized CNN architectures

- Hyperparameter tuning and final training

- Explainability analysis using Grad-CAM and Grad-CAM++

As shown in Fig 3, the proposed framework integrates transformer baselines and customized CNN architectures within a unified training and evaluation pipeline.

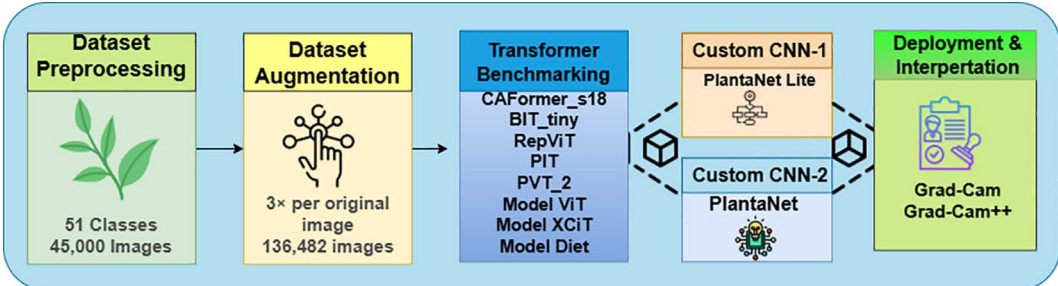

**Fig 3. Overview of the proposed end-to-end pipeline for multi-crop plant disease classification, including dataset construction, augmentation, transformer benchmarking, customized CNN training, evaluation, and explainability using Grad-CAM and Grad-CAM++.**

**Algorithm 1: Training and evaluation pipeline for PlantaNet and PlantaNetLite**

```
 1: Input: 𝒟_raw, stochastic augmentation 𝒜 (training only), resize 160×160, batch B=32
 2: Input: transformer set 𝒯, CNNs {M_lite, M_full}, Adamax
 3: Input: epochs E_T=50, E_C=100, MixUp α, LR η, WD λ
 4: Output: trained models and Grad-CAM/Grad-CAM++ overlays
 5: Clean 𝒟_raw and make stratified split → 𝒟_tr, 𝒟_val, 𝒟_te
 6: Apply stochastic augmentation to training samples to generate additional variants
 7:   Resize/normalize all samples
 8:   for all T ∈ 𝒯 do
 9:     for e=1 to E_T do
10:       for all mini-batches (x,y) in 𝒟_tr^aug do
11:         p ← T(x)
12:         ℒ ← CE(p, y); update T
13:       end for
14:       validate on 𝒟_val; keep best checkpoint
15:     end for
16:   end for
17:   Select best (η, λ, α) using 𝒟_val
18:   for all M ∈ {M_lite, M_full} do
19:     for e=1 to E_C do
20:       for all mini-batches (x,y) in 𝒟_tr^aug do
21:         (x̃, ỹ) ← MixUp(x, y; α)
22:         p ← M(x̃)
23:         ℒ ← CE(p, ỹ); update M
24:       end for
25:       validate on 𝒟_val; keep best checkpoint
26:     end for
27:     test best checkpoint on 𝒟_te; report metrics
28:   end for
29:   Generate Grad-CAM / Grad-CAM++ overlays for selected test samples
```

This modular design ensures reproducibility, fair model comparison, and a clear eparation between baseline analysis and proposed architectural innovations. The overall training and evaluation procedure is summarized in Algorithm 5.1.

### 3.2 Dataset construction and class distribution

A large-scale multi-crop plant disease dataset was constructed by aggregating original (non-augmented) RGB leaf images from multiple publicly available repositories commonly used in agricultural computer vision research [18]. The initial raw collection consisted of approximately 45,000 images spanning both diseased and healthy leaves.

All images were manually verified to remove low-quality samples and duplicates. Each verified image was assigned to one of 51 classes, representing disease categories and healthy states across 12 crop species: corn, rice, tomato, bean, potato, banana, grape, apple, mango, strawberry, blueberry, and bell pepper. The overall class distribution across all 51 classes and dataset splits is summarized in Fig 2.

After manual verification, the dataset was first partitioned using class-wise stratified splitting into training (70%), validation (15%), and test (15%) subsets with a fixed random seed (42) to ensure reproducibility and prevent data leakage. Augmentation was applied strictly to the training subset only. The validation and test subsets were not augmented, except for deterministic resizing and normalization.

Following augmentation of the training subset, the total number of samples used during training reached 95,504 images, while the validation and test sets contained 20,472 and 20,506 images, respectively. A summary of the dataset composition, split protocol, and training configuration is reported in Table 2.

Representative visual examples from different crop–disease categories included in the dataset are shown in Fig 4.

**Table 2. Dataset summary and split.**

| Item | Value |
|---|---|
| Total classes | 51 (healthy + diseased) |
| Crops included | corn, rice, tomato, bean, potato, banana, grape, apple, mango, strawberry, blueberry, bell pepper |
| Total original images | ≈45,000 |
| Training images (after augmentation) | 95,504 |
| Validation images | 20,472 |
| Test images | 20,506 |
| Augmentation applied to | Training subset only |
| Image size used | 160×160 |
| Batch size | 32 |

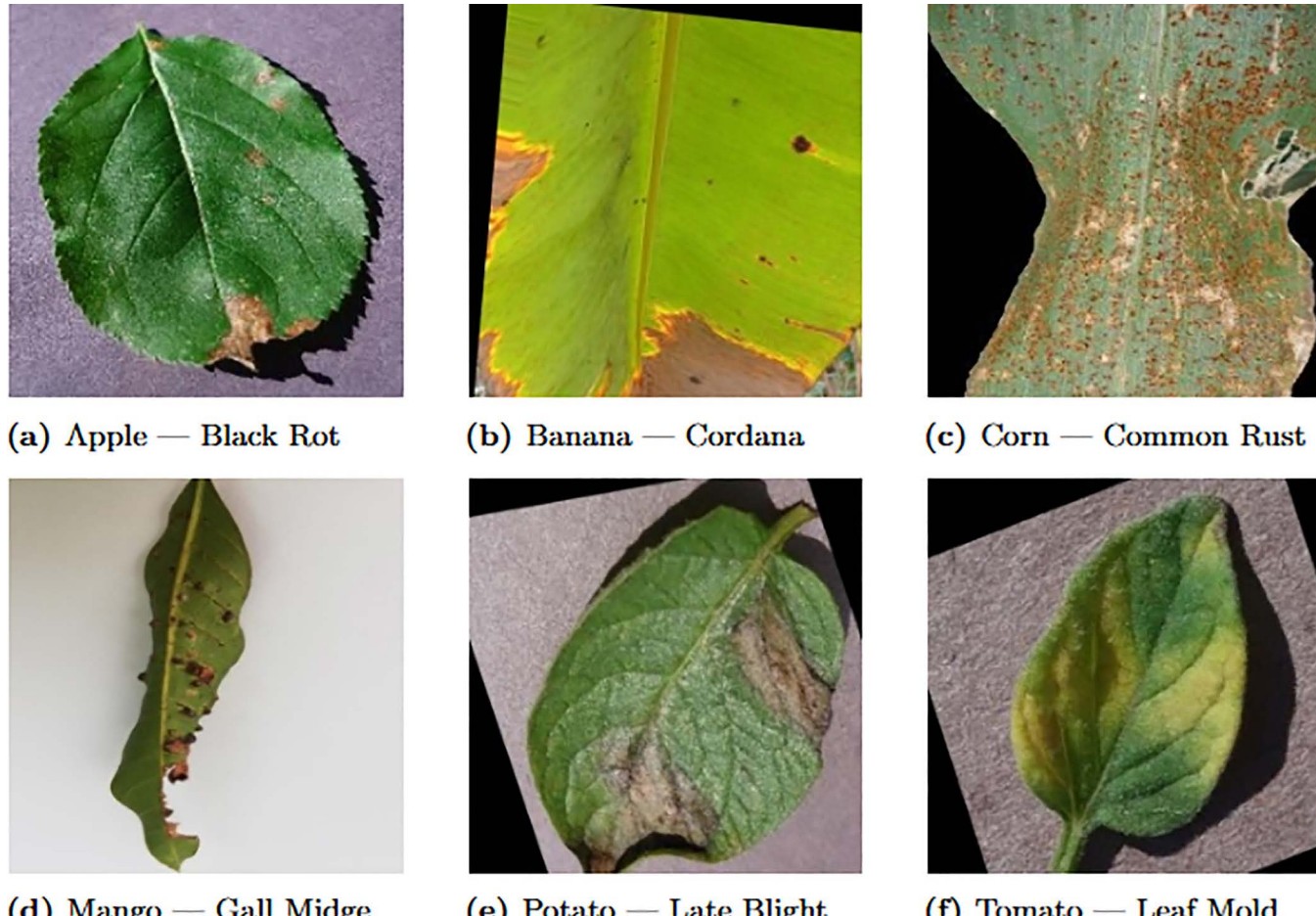

**(a)** Apple — Black Rot  **(b)** Banana — Cordana  **(c)** Corn — Common Rust
**(d)** Mango — Gall Midge  **(e)** Potato — Late Blight  **(f)** Tomato — Leaf Mold

**Fig 4. Representative leaf images from six crop–disease classes included in the proposed multi-crop plant disease dataset. (a)** Apple — Black Rot, **(b)** Banana — Cordana, **(c)** Corn — Common Rust, **(d)** Mango — Gall Midge, **(e)** Potato — Late Blight, **(f)** Tomato — Leaf Mold.

### 3.3 Data augmentation and preprocessing

Data augmentation was applied exclusively to the training subset after stratified splitting to improve robustness against real-world variability such as illumination changes, viewing angles, and background clutter. The augmentation process increased the effective size of the training data while preserving validation and test integrity. Visual examples of the applied augmentation operations are shown in Fig 5.

The applied transformations included:

- *Geometric*: horizontal flip, translation, scaling, and rotation

- *Photometric*: brightness/contrast adjustment and RGB shift

- *Noise/Blur*: Gaussian blur

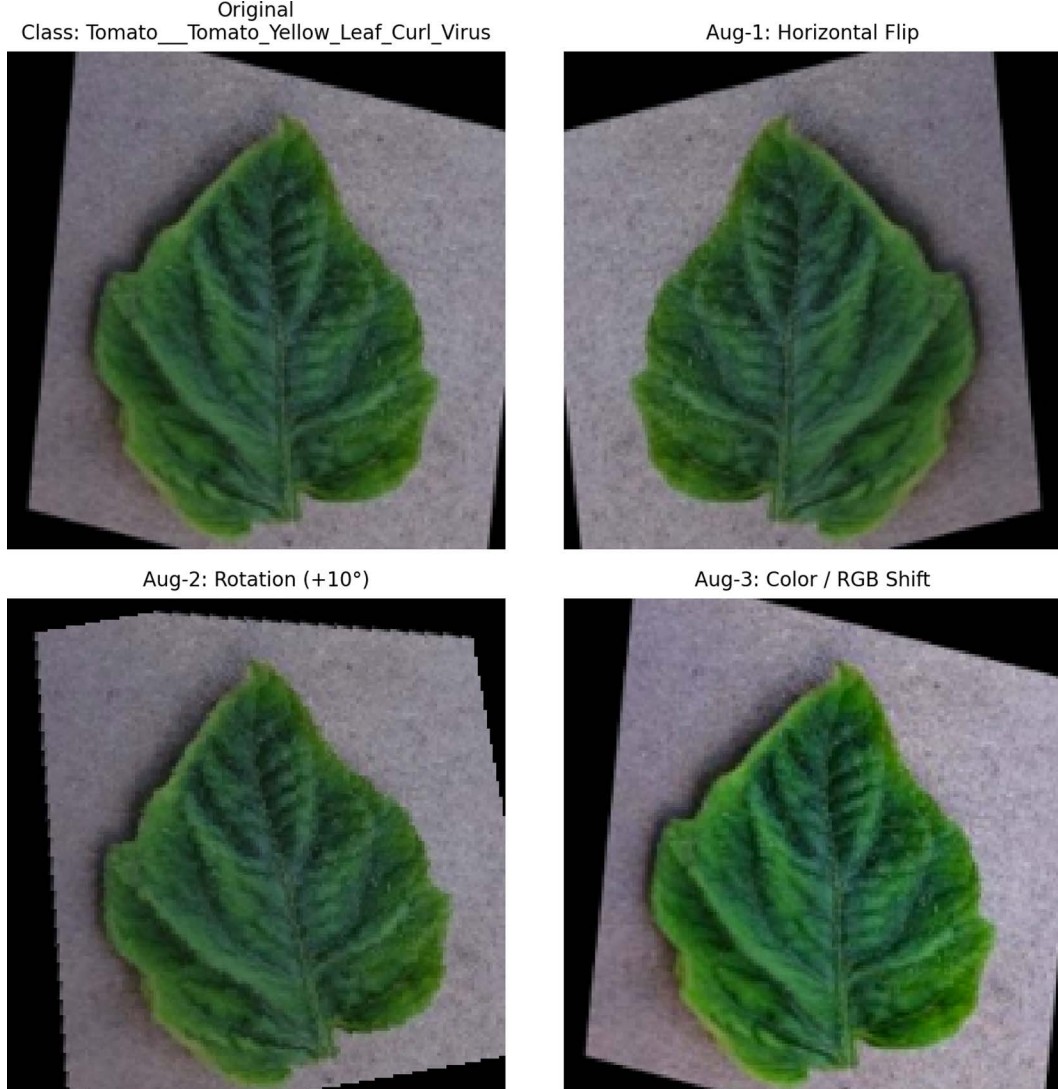

**Fig 5. Visual examples of data augmentation effects, showing an original image followed by three augmented variants generated using the Albumentations pipeline.**

The full augmentation pipeline and parameter ranges are listed in Table 3.

This augmentation strategy preserves disease-specific visual characteristics while introducing realistic variability, improving generalization as recommended in prior plant pathology studies [4,19].

All images were resized to 160 × 160 pixels and normalized using ImageNet mean and standard deviation, a standard practice for transfer learning and stable convergence:

$$\hat{x} = \frac{x - \mu}{\sigma}$$

(1)

where $x$ denotes the original pixel value, and $\mu$ and $\sigma$ represent the channel-wise mean and standard deviation, respectively.

### 3.4 Baseline benchmarking with tiny transformer models

Prior to proposing customized CNN architectures, eight lightweight vision transformer models were benchmarked to establish competitive baselines and analyze the dataset's suitability for attention-based learning. Each transformer was trained for up to 50 epochs using identical preprocessing and optimization settings.

The evaluated models include CAFormer-s18, BiT-Tiny, RepViT, PiT, PVT-v2, ViT-Tiny, XCiT-Tiny, and Deit-Tiny [20–22].

While these models achieved strong validation accuracy, they generally incurred higher computational overhead and longer training times compared to compact CNNs, particularly under resource-constrained settings. These observations motivated the design of efficient, task-specific CNN architectures optimized for plant disease texture recognition.

### 3.5 Customized CNN architectures

Two fully customized convolutional neural networks were introduced:

PlantaNetLite (1.28M parameters): ultra-lightweight model for edge and mobile deployment

PlantaNet (2.58M parameters): higher-capacity model for optimal accuracy–efficiency trade-off

Both models were explicitly designed to capture fine-grained leaf disease patterns, including lesions, blights, discoloration, mildew spread, and edge deformation.

**3.5.1 Depthwise-separable convolution blocks.** To reduce computational cost while maintaining representational power, both architectures employ depthwise-separable convolutions, originally popularized in MobileNet. A standard convolution is factorized into:

**Table 3. Augmentation pipeline.**

| Augmentation | Parameters / Range | Applied To |
|---|---|---|
| Horizontal Flip | $p = 0.5$ | Train only |
| Random Resized Crop | scale (0.9–1.0), ratio (0.95–1.05) | Train only |
| Rotation | ±10° | Train only |
| Brightness/Contrast | ±0.15 | Train only |
| Translation | small random shift (per Albumentations) | Train only |
| Scaling | small random zoom (per Albumentations) | Train only |
| Gaussian Blur | low-strength | Train only |
| RGB Shift | mild channel jitter | Train only |
| Normalization | ImageNet mean/std | Train/Val/Test |

Depthwise convolution: channel-wise spatial filtering

Pointwise convolution (1×1): inter-channel feature mixing

$$C_{std} = k^2 M N H W \qquad (2)$$

$$C_{sep} = k^2 M H W + M N H W \qquad (3)$$

where $k$ is the kernel size, $M$ and $N$ denote the input and output channels, and $H$ and $W$ represent the spatial dimensions of the feature map.

**3.5.2 Group normalization and activation.** Instead of Batch Normalization, Group Normalization (GN) was used to ensure stable training with small batch sizes [15]. Each convolutional block follows:

Dropout was applied after major stages to reduce overfitting.

**3.5.3 PlantaNet Architecture (≈2.58M Parameters).** PlantaNet consists of four main stages:

- **Stem:** Conv$3 \times 3 \to$ GN$\to$ReLU

- **Blocks 1–3:** depthwise-separable convolutions with stride-2 downsampling

- **Extra convolution stage:** Conv$3 \times 3 \to$ GN$\to$ReLU$\to$Dropout

- **Classifier head:** GAP$\to$FC(1024) $\to$ GN$\to$ReLU$\to$Dropout$\to$FC(num_classes)

Channel widths were set to $c_1 = 144$, $c_2 = 224$, $c_3 = 320$, and $c_4 = 448$. The detailed PlantaNet architecture is illustrated in Fig 6.

**3.5.4 PlantaNetLite architecture (≈1.28M parameters).** PlantaNetLite follows the same design philosophy as PlantaNet but employs reduced channel widths and a compact classifier head to satisfy strict memory and latency constraints. The PlantaNetLite architecture is presented in Fig 7.

## 3.6 Training strategy

**3.6.1 Loss function with label smoothing.** Cross-entropy loss with label smoothing was used to prevent overconfident predictions and improve generalization:

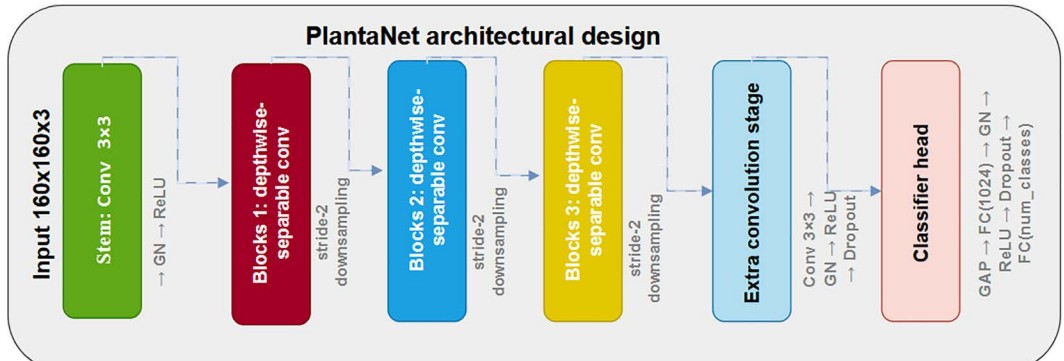

**Fig 6. Detailed architecture of PlantaNet, illustrating depthwise-separable convolution blocks, channel widths, and downsampling stages.**

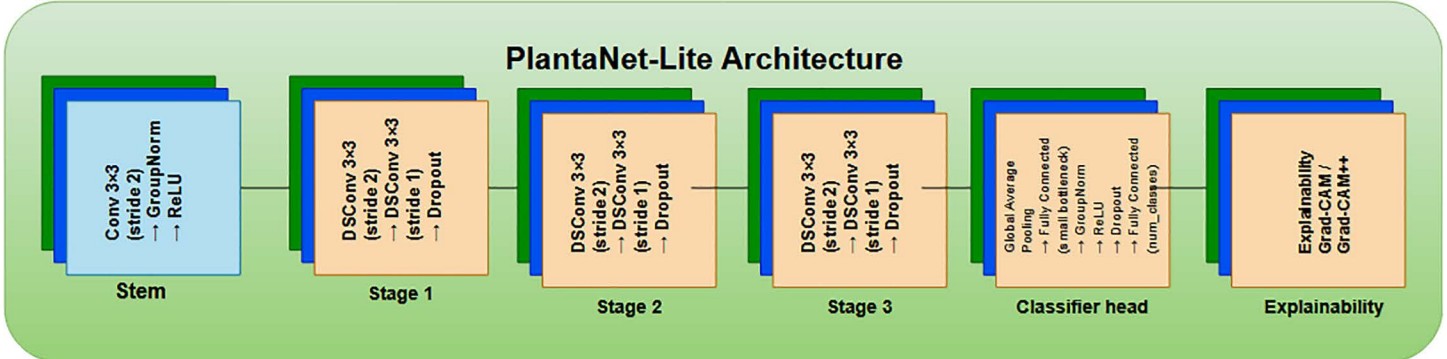

**Fig 7. Architecture of PlantaNetLite, highlighting structural differences compared to PlantaNet and reduced channel widths for lightweight deployment.**

$$L = -\sum_{i=1}^{K} \tilde{y}_i \log(p_i)$$

(4)

where $K$ denotes the number of classes, $p_i$ is the predicted probability for class $i$, and $\tilde{y}_i$ represents the label-smoothed target. Label smoothing coefficient was set to 0.1 for all models.

**3.6.2 MixUp regularization.** MixUp was applied to further improve robustness against occlusion and inter-class similarity [16]:

$$\tilde{x} = \lambda x_a + (1-\lambda)x_b$$

(5)

$$\tilde{y} = \lambda y_a + (1-\lambda)y_b$$

(6)

where $x_a$ and $x_b$ denote input samples with corresponding labels $y_a$ and $y_b$, and $\lambda \sim \text{Beta}(\alpha, \alpha)$ is the MixUp interpolation coefficient.

**3.6.3 Optimization setup.** Training was conducted on a GPU with deterministic seeding. The optimization setup is as follows:

- Optimizer: Adamax [23]

- Initial learning rate: $1 \times 10^{-3}$

- Weight decay: $1 \times 10^{-4}$

- Batch size: 32

- Epochs: 100

- Input resolution: $160 \times 160$

- Random seed: 42

All transformer baselines were initialized with ImageNet-1k pretrained weights and fine-tuned using the same optimizer, learning rate, weight decay, batch size, and preprocessing settings to ensure fair comparison.

## 3.7 Hyperparameter tuning

Hyperparameter tuning was performed on PlantaNet using a grid/random search over the following ranges:

- Learning rate: $\{1 \times 10^{-3}\}, 5 \times 10^{-4}\}$

- Weight decay: $\{1 \times 10^{-4}\}, 5 \times 10^{-4}\}, 1 \times 10^{-5}\}$

- MixUp $\alpha$: $\{0.2, 0.4\}$

  Each configuration was trained for 10 epochs and ranked using validation accuracy and validation loss. The best-performing configuration (Config-3) was:

- LR = $1 \times 10^{-3}$

- Weight decay = $1 \times 10^{-4}$

- MixUp $\alpha$ = 0.2

  The hyperparameter search space used for PlantaNet is summarized in Table 4. A similar tuning process was applied to PlantaNetLite.

## 3.8 Evaluation protocol

Models were evaluated on the held-out test set using:

- Accuracy

- Weighted Precision, Recall, and F1-score

- Confusion matrix

- Training and inference efficiency (parameters, FLOPs, latency, model size)

$$\text{Precision} = \frac{TP}{TP + FP} \tag{7}$$

$$\text{Recall} = \frac{TP}{TP + FN} \tag{8}$$

**Table 4. Hyperparameter search space (PlantaNet).**

| Parameter | Values explored |
|---|---|
| Learning rate (LR) | 1e-3, 5e-4 |
| Weight decay | 1e-4, 5e-4, 1e-5 |
| MixUp $\alpha$ | 0.2, 0.4 |
| Epochs during tuning | 10 |
| Selection criterion | Best validation accuracy (tie-break: val loss stability) |

$$F_1 = \frac{2 \cdot \text{Precision} \cdot \text{Recall}}{\text{Precision} + \text{Recall}} \tag{9}$$

where *TP*, *FP*, and *FN* denote true positives, false positives, and false negatives, respectively, and $F_1$ is the harmonic mean of Precision and Recall.

### 3.9 Explainability using Grad-CAM and Grad-CAM++

To qualitatively examine model attention behavior, Grad-CAM [14] and Grad-CAM++ [11] were applied to representative correctly classified test samples using the best-validation checkpoints.

#### 3.9.1 Grad-CAM.

$$\alpha_k^c = \frac{1}{Z} \sum_i \sum_j \frac{\partial y^c}{\partial A_{ij}^k} \tag{10}$$

$$L_{\text{Grad-CAM}}^c = \text{ReLU}\left( \sum_k \alpha_k^c A^k \right) \tag{11}$$

where $A_{ij}^k$ denotes the activation at spatial location (*i,j*) of the *k*-th feature map in the target convolutional layer, $y^c$ is the score (logit) for class *c*, and *Z* is the number of spatial locations in the feature map (i.e., $Z = H \times W$). The weight $\alpha_k^c$ measures the importance of feature map *k* for class *c*, and $L_{\text{Grad-CAM}}^c$ is the resulting class-discriminative localization map after applying $\text{ReLU}(\cdot)$.

**3.9.2 Grad-CAM++.** Grad-CAM++ incorporates higher-order gradients to localize multiple discriminative regions, which can provide improved localization in cases with multiple or spatially distributed discriminative regions.

### 3.10 Implementation details

All experiments were conducted on Kaggle using NVIDIA Tesla T4 GPUs. Inference latency was measured with batch size 1, averaged over 30 forward passes with CUDA synchronization enabled to ensure accurate timing. All experiments were implemented in PyTorch and executed on Kaggle GPU environments. FLOPs were computed using THOP. Model checkpoints were saved for both best-validation and final-epoch weights, with explainability analyses conducted using best-validation models.

# 4 Results and discussion

This section presents a comprehensive evaluation of all models trained on the proposed multi-crop plant disease dataset comprising approximately 45,000 original images across 51 classes. After stratified splitting and training-only augmentation, the effective number of training samples reached 95,504 images, while validation and test sets contained 20,472 and 20,506 samples, respectively.

### 4.1 Experimental setting and metrics

The dataset was divided into 95,504 training images, 20,472 validation images, and 20,506 test images, maintaining label consistency over all 51 classes. Models were optimized using a classification objective with label smoothing and MixUp for CNN training [16]. To quantify performance, we report training loss, validation loss, training accuracy, validation accuracy,

and for final evaluation, test accuracy, weighted precision, weighted recall, and weighted F1-score. Weighted metrics are reported because the dataset contains multiple crops and diseases with naturally uneven class presence.

## 4.2 Benchmark results of tiny vision transformer models

Eight compact transformer variants were trained for up to 50 epochs to establish strong attention-based baselines [20–22]. The results show consistently high performance across models, demonstrating that the dataset supports strong generalization when trained on diverse augmented samples.

Among transformer baselines:

CAFormer-s18 achieved the highest validation accuracy among transformer baselines (99.89%), with low training and validation losses, indicating stable convergence under the selected optimization protocol [24].

Model XCiT-Tiny also performed near saturation, delivering 99.98% training accuracy and 99.77% validation accuracy, with low training and validation losses, indicating effective convergence under the chosen optimization settings [25].

Model ViT-Tiny achieved 100% training accuracy and 99.697% validation accuracy, indicating rapid convergence, likely due to strong inductive priors under a large dataset [5].

BIT-Tiny remained competitive (99.8% training accuracy, 99.5% validation accuracy), while RepViT, PiT, and PVT-2 produced slightly lower validation accuracies around 97.5–97.9%, reflecting their compact capacity limits for certain visually similar diseases [22,26,27].

Overall, transformer results indicate that attention-based architectures can effectively model disease texture and inter-crop discriminative cues [5,20]. However, transformers usually require more compute per parameter and often exhibit heavier inference pipelines than small CNNs, motivating the second stage of this work. Quantitative performance and training efficiency comparisons are summarized in Table 5.

As shown in Fig 8, the proposed custom CNN models (PlantaNet and PlantaNetLite) achieve competitive validation accuracy compared to the transformer-based baselines under identical training settings. In particular, PlantaNet attains 99.37% validation accuracy, closely matching strong transformer models such as BiT-Tiny, while PlantaNetLite maintains a comparable performance of 99.22%, demonstrating an effective accuracy–efficiency trade-off.

## 4.3 Performance of customized CNNs

### 4.3.1 PlantaNetLite (1.28M Parameters).
PlantaNetLite was designed as a compact CNN emphasizing deployment feasibility while still leveraging multi-scale lesion cues. After hyperparameter tuning and 100 epochs of training, PlantaNetLite achieved:

**Table 5. Performance and training efficiency comparison of transformer baselines and proposed CNN models.**

| Model | Train Loss | Val Loss | Train Acc (%) | Val Acc (%) | Epoch Time (s) | Epochs |
|---|---|---|---|---|---|---|
| CAFormer-s18 | 0.0179 | 0.0023 | 99.40 | 99.89 | 493.26 | 50 |
| BIT-Tiny | 0.0025 | 0.0105 | 99.80 | 99.50 | 474.73 | 50 |
| RepViT | 0.0019 | 0.0083 | 99.80 | 97.50 | 606.98 | 50 |
| PiT | 0.0343 | 0.0612 | 98.82 | 97.92 | 478.50 | 50 |
| PVT-2 | 0.0438 | 0.0611 | 98.02 | 97.74 | 722.50 | 50 |
| ViT-Tiny | 9.06e-05 | 0.01066 | 100.00 | 99.697 | 240.09 | 50 |
| XCiT-Tiny | 0.00095 | 0.00853 | 99.98 | 99.77 | 406.00 | 50 |
| DeiT-Tiny | 0.0013 | 0.0220 | 99.98 | 99.32 | 295.00 | 50 |
| PlantaNet (2.58M) | 0.010 | 0.022 | 99.70 | 99.37 | 415.93 | 100 |
| PlantaNetLite (1.28M) | 0.08 | 0.02 | 97.30 | 99.20 | 235.47 | 100 |

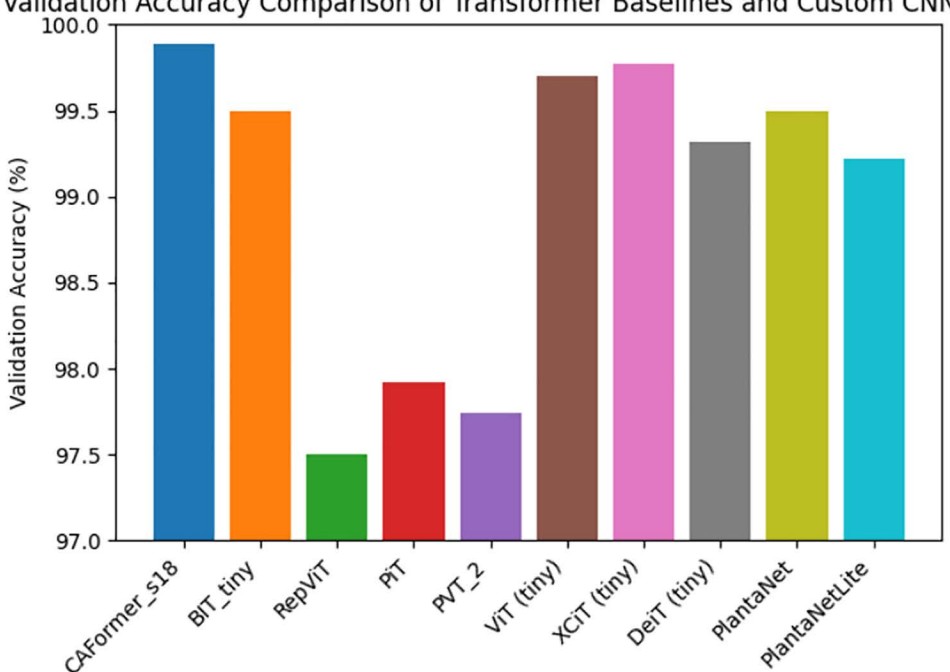

Validation Accuracy Comparison of Transformer Baselines and Custom CNN Models

**Fig 8. Validation accuracy comparison of transformer-based baseline models and the proposed custom CNN architectures (PlantaNet and PlantaNetLite).**

- Best training accuracy: 97.38%

- Best validation accuracy: 99.22%

- Best training loss: 0.0803

- Best validation loss: 0.0232

These results are notable given the small parameter budget (~1.28M). Importantly, the validation accuracy surpasses the training accuracy, which is consistent with strong augmentation and MixUp regularization and indicates robust generalization rather than overfitting [10,16]. The training logs show frequent best-checkpoint updates during early and mid-epochs followed by smooth saturation, confirming stable convergence. Training and validation curves for PlantaNetLite are shown in Fig 9. Final training outcomes for PlantaNetLite are summarized in Table 6.

**Discussion:** PlantaNetLite demonstrates that careful architectural bias (convolutions tuned to lesion patterns), coupled with augmentation and MixUp, can provide competitive performance under a lightweight regime [15].

**4.3.2 PlantaNet (2.58M parameters).** PlantaNet was proposed to maximize accuracy while remaining parameter-efficient. A systematic hyperparameter search selected Config-3, defined by:

- Learning rate: $1 \times 10^{-3}$

- Weight decay: $1 \times 10^{-4}$

- MixUp $\alpha$: 0.2

Training this configuration for 100 epochs yielded the following results:

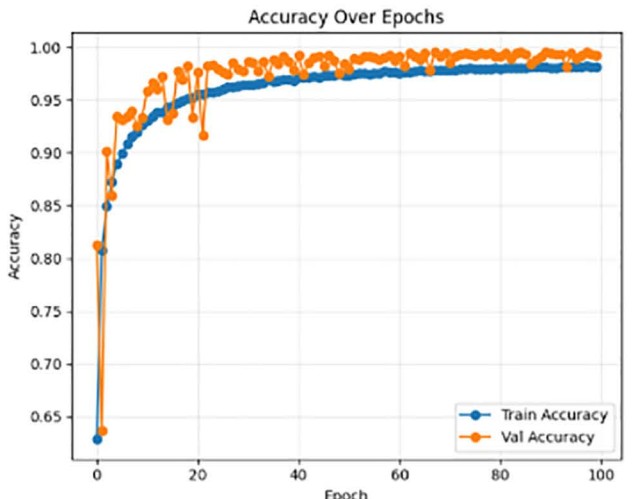
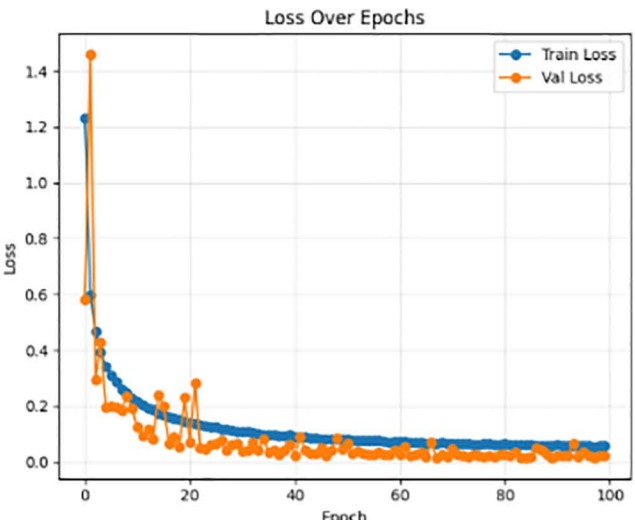

**Fig 9. Training and validation accuracy (left) and loss (right) curves for PlantaNetLite over 100 epochs.**

**Table 6. Final training result summary – PlantaNetLite.**

| Metric | Value |
| --- | --- |
| Epochs | 100 |
| Final Train Acc (%) | 98.14 |
| Final Val Acc (%) | 99.19 |
| Best checkpoint criterion | Highest validation accuracy |

- Validation accuracy: 99.37%

- Test accuracy: 99.66%

- Weighted precision: 0.997

- Weighted recall: 0.997

- Weighted F1-score: 0.997

- Parameters: 2.58M

- FLOPs: 1213.9M

- Inference time: 23.48 ms

- Model size: 9.85 MB

PlantaNet achieved competitive validation performance while maintaining a compact parameter budget. The training history demonstrates rapid early improvement, with accuracy approaching 0.97 within the first few epochs, followed by long-term stabilization around 0.995–0.997. The loss curve decreased sharply during early training and converged to low values (approximately 0.010 training loss and 0.022 validation loss at convergence), indicating stable optimization. The final performance and efficiency summary for PlantaNet is reported in Table 7. Training and validation curves for Planta-Net are shown in Fig 10.

**Table 7. Efficiency summary of PlantaNet.**

| Metric | Value |
|---|---|
| Selected Config | 3 |
| Learning Rate | 1e-3 |
| Weight Decay | 1e-4 |
| MixUp $\alpha$ | 0.2 |
| Epochs | 100 |
| Best Val Acc (%) | 99.37 |
| Test Acc (%) | 99.66 |
| Precision (weighted) | 0.997 |
| Recall (weighted) | 0.997 |
| F1-score (weighted) | 0.997 |

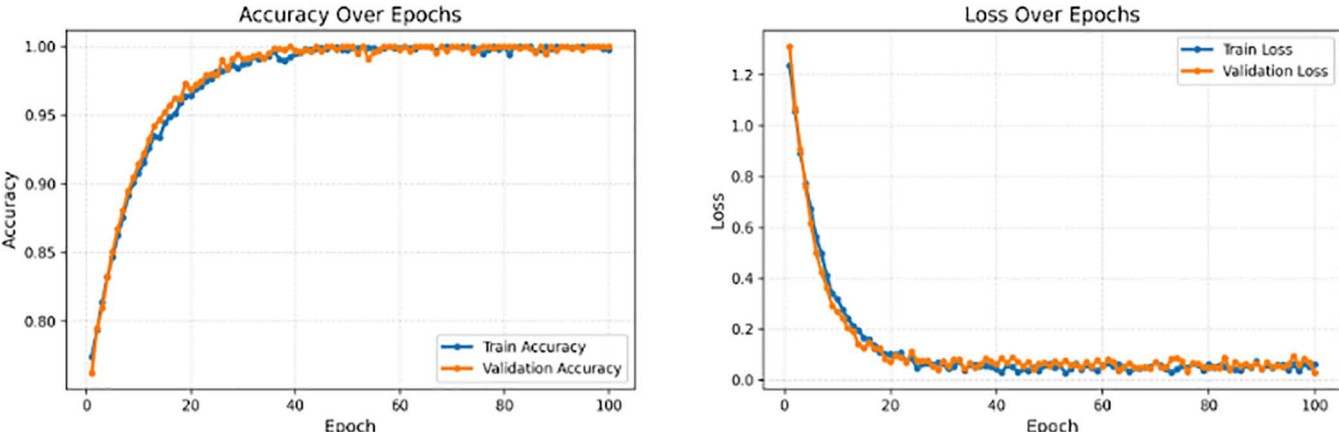

**Fig 10. Training and validation accuracy (left) and loss (right) curves for PlantaNet over 100 epochs.**

**Discussion:** PlantaNet demonstrates that carefully designed convolutional architectures can achieve competitive accuracy while maintaining parameter efficiency, providing a favorable accuracy–efficiency balance under the evaluated experimental settings.

*Accuracy with compactness:* PlantaNet matched or exceeded top transformer accuracies using only 2.58M parameters, indicating that leaf disease recognition benefits from convolutional inductive bias [4,19].

*Efficiency considerations:* With 2.58M parameters and moderate computational cost, PlantaNet maintains a favorable balance between accuracy and model complexity under the evaluated experimental settings.

## 4.4 Hyperparameter tuning impact

Hyperparameter search results for PlantaNet reveal a clear dependency on regularization and data mixing strength [16]. Config-3 (MixUp = 0.2) produced the best validation accuracy during tuning (0.9565) and best loss profile, while heavier mixing (MixUp = 0.4) slightly degraded performance by over smoothing subtle lesion boundaries. These values correspond to the 10-epoch tuning phase and do not represent the final 100-epoch performance. Hyperparameter tuning outcomes are reported in Table 8.

**Table 8. Hyperparameter tuning results for PlantaNet.**

| Config ID | LR | Weight Decay | MixUp $\alpha$ | Best Val Acc | Best Val Loss |
|---|---|---|---|---|---|
| 3 | 1e-3 | 1e-4 | 0.2 | 0.956526 | 0.873800 |
| 6 | 1e-3 | 5e-4 | 0.4 | 0.956428 | 0.871576 |
| 5 | 1e-3 | 5e-4 | 0.2 | 0.955451 | 0.862726 |
| 1 | 1e-3 | 1e-5 | 0.2 | 0.950274 | 0.886800 |
| 2 | 1e-3 | 1e-5 | 0.4 | 0.947343 | 0.936944 |
| 4 | 1e-3 | 1e-4 | 0.4 | 0.944900 | 0.913717 |
| 11 | 5e-4 | 5e-4 | 0.2 | 0.931321 | 0.956827 |
| 9 | 5e-4 | 1e-4 | 0.2 | 0.930832 | 0.949446 |
| 8 | 5e-4 | 1e-5 | 0.4 | 0.918914 | 1.000900 |
| 12 | 5e-4 | 5e-4 | 0.4 | 0.917790 | 1.007759 |

## 4.5 Ablation study

To assess the contribution of key architectural components in PlantaNet, controlled ablation experiments were conducted by modifying one component at a time while keeping all other training settings fixed. The results are summarized in Table 9.

The ablation focuses specifically on architectural design elements, including normalization strategy, dropout regularization, and the additional convolutional refinement block. Training hyperparameters such as MixUp and label smoothing were kept unchanged during this analysis.

Replacing Group Normalization with Batch Normalization resulted in a modest reduction in validation accuracy (99.37% → 98.90%), indicating improved stability of GroupNorm under the selected batch size (32). Removing dropout produced a slight decrease in validation accuracy (99.37% → 99.11%), suggesting its regularization benefit. Excluding the additional convolutional refinement block led to a minor reduction (99.37% → 99.28%), indicating that its contribution is incremental rather than dominant.

Overall, these architectural ablation results provide empirical support for the selected normalization and regularization strategy.

## 4.6 Explainability via Grad-CAM and Grad-CAM++

Explainability was evaluated using both Grad-CAM and Grad-CAM++ on one correctly classified test sample per class, using the best validation weights of PlantaNet to ensure interpretation aligns with the most generalizable model state [11,14].

Results show that:

Qualitative inspection of Grad-CAM and Grad-CAM++ visualizations suggests that model activations frequently overlap with visible disease-affected regions in representative samples. Healthy samples typically produced more diffuse activation patterns across leaf surfaces. Grad-CAM++ generally yielded more localized saliency maps compared to Grad-CAM. The explainability settings and outputs for each model are summarized in Table 10.

Fig 11 compares the localization behavior of PlantaNet and PlantaNetLite for two representative diseases, demonstrating consistent focus on biologically meaningful lesion regions.

Discussion: The qualitative visualizations provide insight into model attention behavior, although a quantitative localization assessment is beyond the scope of this study.

## 4.7 Overall comparative analysis and key findings

The experimental results indicate that both transformer-based baselines and customized CNN architectures achieve high validation accuracy on the proposed multi-crop dataset. While transformer models establish strong attention-based baselines, the proposed CNNs demonstrate competitive performance with substantially fewer parameters.

**Table 9. Architectural ablation study on PlantaNet (validation accuracy).**

| Configuration | Validation Accuracy (%) |
|---|---|
| Full model (GroupNorm + Dropout + Extra Conv) | 99.37 |
| BatchNorm instead of GroupNorm | 98.90 |
| Without Dropout | 99.11 |
| Without Extra Convolution Block | 99.28 |

**Table 10. Explainability output inventory.**

| Model | Explainability Method | Target Layer | Samples Used | Output |
|---|---|---|---|---|
| PlantaNetLite | Grad-CAM | last conv block | 1 correct sample/class | heatmaps + overlays |
| PlantaNetLite | Grad-CAM++ | last conv block | 1 correct sample/class | sharper saliency maps |
| PlantaNet | Grad-CAM | conv_extra[0] | 1 correct sample/class | heatmaps + overlays |
| PlantaNet | Grad-CAM++ | conv_extra[0] | 1 correct sample/class | refined localization |

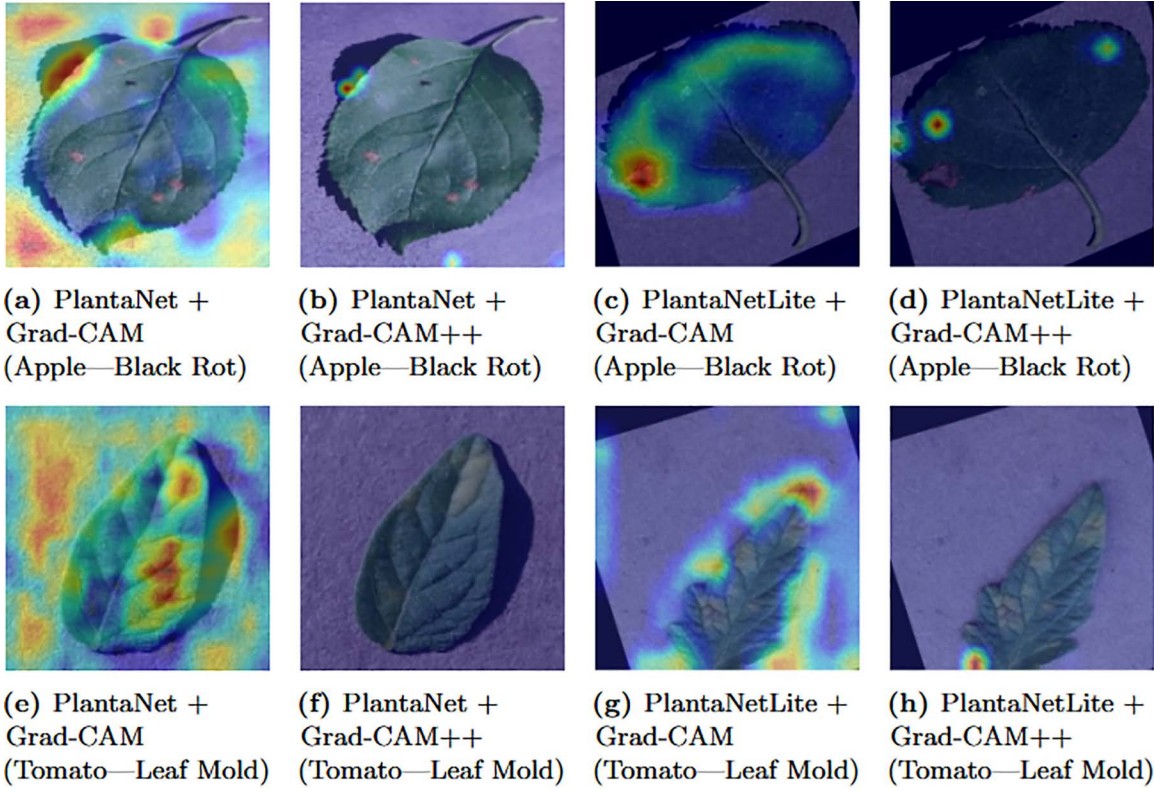

**(a)** PlantaNet + Grad-CAM (Apple—Black Rot) **(b)** PlantaNet + Grad-CAM++ (Apple—Black Rot) **(c)** PlantaNetLite + Grad-CAM (Apple—Black Rot) **(d)** PlantaNetLite + Grad-CAM++ (Apple—Black Rot)

**(e)** PlantaNet + Grad-CAM (Tomato—Leaf Mold) **(f)** PlantaNet + Grad-CAM++ (Tomato—Leaf Mold) **(g)** PlantaNetLite + Grad-CAM (Tomato—Leaf Mold) **(h)** PlantaNetLite + Grad-CAM++ (Tomato—Leaf Mold)

**Fig 11. Comparison of Grad-CAM and Grad-CAM++ explanations for correctly classified test samples from two representative disease classes using PlantaNet and PlantaNetLite. (a)** PlantaNet + Grad-CAM (Apple—Black Rot), **(b)** PlantaNet + Grad-CAM++ (Apple—Black Rot), **(c)** PlantaNetLite + Grad-CAM (Apple—Black Rot), **(d)** PlantaNetLite + Grad-CAM++ (Apple—Black Rot), **(e)** PlantaNet + Grad-CAM (Tomato—Leaf Mold), **(f)** PlantaNet + Grad-CAM++ (Tomato—Leaf Mold), **(g)** PlantaNetLite + Grad-CAM (Tomato—Leaf Mold), **(h)** PlantaNetLite + Grad-CAM++ (Tomato—Leaf Mold).

PlantaNet and PlantaNetLite highlight that convolutional inductive bias remains effective for fine-grained plant disease recognition under constrained parameter budgets. The qualitative explainability analysis provides additional insight into model attention patterns across representative disease categories.

Further evaluation on independently collected field datasets would provide a more comprehensive assessment of real-world robustness.

# 5 Conclusion, limitations, and future work

## 5.1 Conclusion

This study presented a comprehensive and explainable plant disease classification framework covering 51 disease/healthy categories across diverse crops using a dataset aggregated from multiple public repositories comprising approximately 45,000 original images across 51 classes; after stratified splitting, training-only augmentation increased the effective training set to 95,504 images, with 20,472 validation and 20,506 test samples. A three-stage evaluation was conducted: (1) benchmarking eight compact transformer-based models, (2) proposing and optimizing two fully customized lightweight CNNs—PlantaNetLite (1.28M parameters) and PlantaNet (2.58M parameters), and (3) validating interpretability using Grad-CAM and Grad-CAM++.

The transformer baselines trained for up to 50 epochs achieved consistently high validation performance. In particular, CAFormer-s18 reached 99.89% validation accuracy, indicating that attention-based models can learn fine-grained leaf pathology cues in a high-diversity dataset [20]. However, the proposed CNNs achieved performance competitive with transformer baselines while using substantially fewer parameters and maintaining a lighter deployment footprint.

PlantaNet achieved strong performance, reaching 99.37% validation accuracy and 99.66% test accuracy, with weighted Precision/Recall/F1 of 0.997, while using only 2.58M parameters. Its efficiency profile is suitable for resource-constrained deployment, requiring 1213.9M FLOPs, 23.48 ms inference time, and 9.85 MB storage. These results show that a carefully designed CNN with depthwise-separable blocks, GroupNorm stabilization, and MixUp-based regularization can achieve performance competitive with compact transformer baselines under the evaluated experimental settings [15].

PlantaNetLite was designed for ultra-lightweight scenarios and achieved 99.22% validation accuracy with only 1.28M parameters, demonstrating that strong plant disease recognition is feasible under strict computational constraints.

Finally, qualitative Grad-CAM and Grad-CAM++ visualizations suggest that PlantaNet and PlantaNetLite often focus on symptom-relevant regions rather than obvious background artifacts. These activation maps frequently highlight characteristic disease patterns such as lesion clusters, necrotic textures, rust spots, and mosaic discolorations, providing qualitative interpretability evidence [11,14]. Overall, the results establish that the proposed customized CNNs provide an efficient and explainable alternative to transformer models for multi-crop plant disease recognition. This work targets researchers and practitioners seeking efficient, explainable deep learning solutions for real-world agricultural disease diagnosis under limited computational resources.

## 5.2 Limitations

Although the results are strong, several limitations should be acknowledged:

- **Controlled dataset bias:** While the dataset is large and diverse, many images originate from curated or semi-controlled sources. Real field images may include harsher illumination changes, partial occlusion, overlapping leaves, mixed infections, soil artifacts, and motion blur, which may reduce performance under uncontrolled conditions [1,2].

- **Single-label assumption:** Each image is assigned a single dominant class. In practice, co-infections and multiple stress factors may occur simultaneously, which is not explicitly modeled in this work.

- **Class-level interpretability:** Grad-CAM/Grad-CAM++ provide qualitative localization but do not produce pixel-level lesion segmentation or quantitative severity estimates [11,14].

- **No external cross-dataset validation:** The models were evaluated on a fixed split of one unified dataset. Testing on external benchmarks or region-specific field datasets would strengthen evidence of generalization [1]. **Limited explainability sampling:** Explainability was illustrated using one correctly classified sample per class, which may be sensitive to sample selection and does not support statistical claims.

### 5.3  Future Work

Future work will explore the following directions:

- **Field-domain adaptation and robustness testing:** Evaluate the proposed models on farm-captured images under domain shift, potentially using domain adaptation, test-time augmentation, or robust training strategies [1,2].

- **Multi-label and co-infection recognition:** Extend the formulation to multi-label prediction to better reflect real agricultural scenarios.

- **Lesion segmentation and severity estimation:** Integrate a lightweight segmentation head or multi-task learning to support both disease identification and severity estimation.

- **Edge-AI and mobile deployment:** Investigate quantization-aware training and hardware-aware optimization for deployment on mobile and edge devices.

- **Explainability beyond Grad-CAM:** Combine Grad-CAM++ with attribution methods such as Integrated Gradients to provide complementary explanations and stronger trust evidence.

## Author contributions

**Data curation:** Md. Sifat Haque Zidan, Md. Kayes Mia.

**Formal analysis:** Md. Kayes Mia.

**Investigation:** Md. Sifat Haque Zidan, Ahmed Faizul Haque Dhrubo.

**Supervision:** Ahmed Faizul Haque Dhrubo, Mohammad Abdul Qayum.

**Validation:** Al Imran, Muhammad Hussain.

**Visualization:** Al Imran, Muhammad Hussain.

**Writing – original draft:** Ahmed Faizul Haque Dhrubo.

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
