## [Decision Letter · Decision Letter 0]

28 Jan 2026

PlantaNet and PlantaNetLite: Efficient and Explainable Multi-Crop Plant Disease Classification via Transformer Benchmarking and Custom Lightweight CNNs

PLOS One

Dear Dr. Dhrubo,

Thank you for submitting your manuscript to PLOS ONE. After careful consideration, we feel that it has merit but does not fully meet PLOS ONE’s publication criteria as it currently stands. Therefore, we invite you to submit a revised version of the manuscript that addresses the points raised during the review process.

We look forward to receiving your revised manuscript.

Kind regards,

Marco Antonio Moreno-Armendariz, Ph.D.

Academic Editor

PLOS One

**Journal Requirements:**

3. Please ensure that you refer to Figures 2, 5, 6, 7, 9,  and 10, in your text as, if accepted, production will need this reference to link the reader to the figure.

4. Please upload a copy of Figure 12, to which you refer in your text on page 15. If the figure is no longer to be included as part of the submission please remove all reference to it within the text.

5. We note you have included a table to which you do not refer in the text of your manuscript. Please ensure that you refer to Tables 1, 2, 3, 4, 5, 6, 7, 8, and 9 in your text; if accepted, production will need this reference to link the reader to the Table.

**Additional Editor Comments:**

Dear Author,

Please fill in the reviewer's comments.

Best wishes,

Marco.

Reviewers' comments:

Reviewer's Responses to Questions

**Comments to the Author**

1. Is the manuscript technically sound, and do the data support the conclusions?

Reviewer #1: Yes

Reviewer #2: Partly

2. Has the statistical analysis been performed appropriately and rigorously?

Reviewer #1: N/A

Reviewer #2: No

3. Have the authors made all data underlying the findings in their manuscript fully available?

Reviewer #1: Yes

Reviewer #2: Yes

4. Is the manuscript presented in an intelligible fashion and written in standard English?

Reviewer #1: Yes

Reviewer #2: Yes

Reviewer #2: This manuscript proposes lightweight CNN-based models (PlantaNet and PlantaNetLite) for multi-crop plant disease classification and presents corresponding experimental results. While the model design is generally well motivated and the architectural choices are reasonable, the paper falls short of the level of rigor expected for PLOS ONE, particularly in terms of dataset description, experimental analysis, and clarity of contribution. Several methodological and organizational issues further weaken the manuscript. In its current form, the paper does not provide sufficient evidence or insight to justify its claims beyond performance comparisons, and substantial revision would be required to improve its clarity and scientific value.

## Strengths

S1. The proposed models are built from established components, and the authors provide reasonable and generally clear explanations for why each architectural choice (e.g., lightweight convolutions, normalization strategy) is appropriate for the target problem.

S2. The study demonstrates that it is possible to reduce the number of model parameters while maintaining, or even improving, generalization performance, which is relevant for resource-constrained deployment scenarios.

## Weaknesses

W1. The manuscript repeatedly refers to a dataset size of 136,482 images; however, this count includes images generated through data augmentation. In standard practice, dataset size should be reported based on the number of original samples, with augmentation described separately as part of the training strategy. Reporting the augmented count as the dataset size is misleading and inflates the apparent scale of the data.

W2. The paper states that the dataset consists of 51 classes and that stratified splitting was performed, but critical details are missing:

- What exactly does each of the 51 classes represent?

- Was stratification performed at the level of original images, prior to augmentation?

- How was it ensured that augmented versions of the same original image did not appear across training, validation, and test splits?

Without explicit clarification, the possibility of data leakage cannot be ruled out, which would seriously compromise the validity of the reported results.

W3. PLOS ONE requires rigorous and transparent data analysis, but the dataset description in this manuscript is superficial. Missing information includes:

- The number and proportion of healthy vs. diseased samples.

- The distribution of samples across plant species.

- Disease/healthy ratios within each plant species.

Given the large number of classes, such statistics are essential for evaluating dataset balance and interpreting model performance.

W4. The structure of the Methodology section requires significant revision.

- Benchmarking against ViT-based models is experimental evaluation, not methodology, and is out of place in this section.

- Baseline comparisons should be described in the experimental or results section.

- The use of approximate values (e.g., "≈20,506 validation images") in Table 2 is inappropriate; exact numbers should be reported.

W5. Grad-CAM and Grad-CAM++ are post-hoc qualitative analysis tools, yet they are introduced as part of the core methodology. Their role is better framed as an evaluation or analysis method in the experimental section. (Although details are later provided in Section 0.16, mentioning them as part of the methodology at the outset is misleading and should be avoided.)

W6. Algorithm 1 presents a generic training loop with no distinctive or novel elements. As an audience, including such pseudocode does not meaningfully aid understanding. A concise schematic or flow diagram summarizing the overall pipeline would provide a more effective overview.

W7. Several key components are mentioned but not explained with sufficient technical rigor:

- Group Normalization is introduced without adequate theoretical or practical justification.

- The motivation for using both Grad-CAM and Grad-CAM++ is unclear. The manuscript does not clearly articulate what additional insight Grad-CAM++ provides over Grad-CAM, nor does it draw strong conclusions from the explainability results.

W8. The experimental evaluation largely stops at performance comparison between models. No ablation studies are provided to assess the impact of data augmentation, normalization choices, or architectural components. Qualitative analysis of individual decision cases is minimal. Grad-CAM/Grad-CAM++ analysis is based on a very limited number of samples, which weakens the credibility of the interpretability claims.

W9. Despite extensive experimental results, no statistical significance testing is reported. Also, the paper does not provide data-centric analyses that could yield deeper insight into model behavior or failure modes. Given the emphasis on empirical evaluation, these represent a significant weakness.

W10. Although lightweight deployment is a central motivation, claims regarding suitability for edge devices remain superficial. Even limited real-device experiments (e.g., latency on an actual edge device, quantization tests) would substantially strengthen the argument.

## Minor suggestions

1. While reporting hyperparameters is useful, presenting them as long bullet-point lists is inefficient; a more compact or tabular presentation would improve readability.

2. It is not clear what message Table 8 is intended to convey. The table occupies considerable space but does not lead to a clear insight or conclusion. (More generally, the manuscript devotes excessive attention to hyperparameter search, while providing little analysis of how the model responds to the data itself.)

3. Subsection numbering should be revised (e.g., 0.1, 0.2, 0.3 is unconventional and confusing).

4. All baseline models should be introduced with their full names at first mention, with citations placed immediately after the model names (e.g., PVT, PiT, XCiT, RepViT, CAFormer, etc.).

.

Reviewer #2: No

---

## [Author Response · Author response to Decision Letter 1]

16 Mar 2026

Dear Editor,

According to all reviewers comment I corrected in my main manuscript and highlight them in blue color and I also upload a doc file according to reviewer comment which is approximately 20 questions.

Sincerely,

Ahmed Faizul Haque Dhrubo

Research Assitant

North South University

Dhaka, Bangladesh

---

## [Decision Letter · Decision Letter 1]

27 Mar 2026

PlantaNet and PlantaNetLite: Efficient and Explainable Multi-Crop Plant Disease Classification via Transformer Benchmarking and Custom Lightweight CNNs

PONE-D-25-69040R1

Dear Dr. Dhrubo,

We’re pleased to inform you that your manuscript has been judged scientifically suitable for publication and will be formally accepted for publication once it meets all outstanding technical requirements.

Kind regards,

Marco Antonio Moreno-Armendariz, Ph.D.

Academic Editor

PLOS One

Additional Editor Comments (optional):

Reviewers' comments:

Reviewer's Responses to Questions

**Comments to the Author**

Reviewer #1: All comments have been addressed

2. Is the manuscript technically sound, and do the data support the conclusions?

Reviewer #1: Yes

3. Has the statistical analysis been performed appropriately and rigorously?

Reviewer #1: N/A

4. Have the authors made all data underlying the findings in their manuscript fully available?

Reviewer #1: No

5. Is the manuscript presented in an intelligible fashion and written in standard English?

Reviewer #1: Yes

Reviewer #1: Please revise the GPT-generated contents.

.

Reviewer #1: No

---

## [Editor Report · Acceptance letter]

PONE-D-25-69040R1

PLOS One

Dear Dr. Dhrubo,

I'm pleased to inform you that your manuscript has been deemed suitable for publication in PLOS One. Congratulations! Your manuscript is now being handed over to our production team.

Kind regards,

on behalf of

Professor Marco Antonio Moreno-Armendariz

Academic Editor

PLOS One